# Partial restoration of immune response in Hepatitis C patients after viral clearance by direct-acting antiviral therapy

**Meritxell Llorens-Revull**[1,2,3], **Maria Isabel Costafreda**[1,4], **Angie Rico**[1,2,4], **Mercedes Guerrero-Murillo**[1], **Maria Eugenia Soria**[5,6], **Sofía Píriz-Ruzo**[1], **Elena Vargas-Accarino**[1], **Pablo Gabriel-Medina**[7], **Francisco Rodríguez-Frías**[2,3,7,8], **Mar Riveiro-Barciela**[2,9], **Celia Perales**[2,5,6], **Josep Quer**[1,2,3], **Silvia Sauleda**[2,4], **Juan Ignacio Esteban**[1,2,3,9], **Marta Bes**[2,4]*

1 Liver Diseases-Viral Hepatitis Laboratory, Liver Unit, Vall d'Hebron Institut de Recerca (VHIR), Hospital Universitari Vall d'Hebron (HUVH), Vall d'Hebron Barcelona Hospital Campus, Barcelona, Spain, 2 Centro de Investigación Biomédica en Red de Enfermedades Hepáticas y Digestivas (CIBERehd), Instituto de Salud Carlos III, Madrid, Spain, 3 Universitat Autònoma de Barcelona (UAB), Barcelona, Spain, 4 Transfusion Safety Laboratory, Banc de Sang i Teixits (BST), Barcelona, Spain, 5 Centro de Biología Molecular "Severo Ochoa" (CSIC-UAM), Campus de Cantoblanco, Madrid, Spain, 6 Department of Clinical Microbiology, IIS-Fundación Jiménez Díaz, UAM, Madrid, Spain, 7 Liver Pathology Laboratory, Biochemistry and Microbiology Unit, Vall d'Hebron Hospital Universitari (HUVH), Vall d'Hebron Barcelona Hospital Campus, Barcelona, Spain, 8 Clinical Biochemistry Unit, Vall d'Hebron Institut de Recerca (VHIR), Hospital Universitari Vall d'Hebron (HUVH), Vall d'Hebron Barcelona Hospital Campus, Barcelona, Spain, 9 Liver Unit, Department of Internal Medicine, Hospital Universitari Vall d'Hebron (HUVH), Vall d'Hebron Barcelona Hospital Campus, Barcelona, Spain

* mbes@bst.cat

**Data Availability Statement:** All relevant data are within the paper and its Supporting Information files.

## Abstract

### Background & aims

HCV CD4+ and CD8+ specific T cells responses are functionally impaired during chronic hepatitis C infection. DAAs therapies eradicate HCV infection in more than 95% of treated patients. However, the impact of HCV elimination on immune responses remain controversial. Here, we aimed to investigate whether HCV cure by DAAs could reverse the impaired immune response to HCV.

### Methods

We analyzed 27 chronic HCV infected patients undergoing DAA treatment in tertiary care hospital, and we determined the phenotypical and functional changes in both HCV CD8+ and CD4+ specific T-cells before and after viral clearance. PD-1, TIM-3 and LAG-3 cell-surface expression was assessed by flow cytometry to determine CD4+ T cell exhaustion. Functional responses to HCV were analyzed by IFN-γ ELISPOT, intracellular cytokine staining (IL-2 and IFN-γ) and CFSE-based proliferation assays.

### Results

We observed a significant decrease in the expression of PD-1 in CD4+ T-cells after 12 weeks of viral clearance in non-cirrhotic patients (p = 0.033) and in treatment-naive patients (p = 0.010), indicating a partial CD4 phenotype restoration. IFN-γ and IL-2 cytokines

**Funding:** This study was funded by Instituto de Salud Carlos III, cofinanced by the European Regional Development Fund (ERDF): grant numbers PI16/00337, PI18/00210 and PI19/00301. C.P. is supported by the Miguel Servet program of the Instituto de Salud Carlos III (CP14/00121 and CPII19/00001) cofinanced by the European Regional Development Fund (ERDF).

**Competing interests:** The authors have declared that no competing interests exist.

production by HCV-specific CD4+ and CD8+ T cells remained impaired upon HCV eradication. Finally, a significant increase of the proliferation capacity of both HCV CD4+ and CD8+ specific T-cells was observed after HCV elimination by DAAs therapies.

## Conclusions

Our results show that in chronically infected patients HCV elimination by DAA treatment lead to partial reversion of CD4+ T cell exhaustion. Moreover, proliferative capacity of HCV-specific CD4+ and CD8+ T cells is recovered after DAA's therapies.

## Introduction

Hepatitis C is an infectious disease caused by HCV, associated with significant liver-related morbidity and mortality. Approximately, 15–45% of infected people spontaneously clear the infection [1], but the vast majority of infections course asymptomatically and become chronic. Of those with chronic HCV infection, 15–45% will silently develop advanced liver fibrosis/cirrhosis within 20–30 years [2] and 2–4% of all HCV cases will develop liver cancer or liver failure [3]. Short-duration therapy with Direct-Acting Antivirals (DAAs), lead to a Sustained Virological Response (SVR) in more than 95% of patients regardless of viral genotype and with minimal side effects [4]. However, individuals with advanced liver disease are still at risk of developing HCC [5] and many of them remain on liver transplant lists [6] after HCV infection cure.

HCV-specific T cell immune responses are required for viral control. HCV-specific CD4 + T cells have important helper functions, contributing to maintain the CD8+ T cell response and preventing viral escape from T cell response [7]. Lack or loss of helper functions by CD4 + T cells can result in a dysfunctional CD8+ T cell response and an immune response failure against the virus [8, 9]. HCV-specific CD8+ T cells play an essential role in HCV control since they may be involved not only in the elimination of infected hepatocytes but also in non-cytolytic effector mechanisms [10, 11]. Self-limiting HCV infections are associated with expansion of virus-specific CD8+ T cells, a broad CD4+ T cell responses, and a strong cytotoxic T lymphocyte response [12–15]. During chronic HCV infection, constant exposure to viral antigens along with immunological factors results in varying degrees of functional impairment of HCV-specific T-cell effector functions, contributing to viral persistence [16–18].

Several groups have investigated whether DAAs treatment, and the subsequent elimination of HCV, can restore the HCV-specific immune responses, although controversial results have been reported. For instance, it has been demonstrated that HCV-specific T cell functions does not completely ameliorate after SVR [19]. In contrast, other studies have observed an increase in T-cell functionality after DAA treatment [20]. It is known that, early impairment of proliferation may contribute to loss of T cell response and chronic HCV persistence [21]. Despite the fact that there is not yet a solid evidence that T cell proliferation is fully recovered after HCV cure, several studies have seen a rise in the proliferative profile of CD8 T cells in chronic HCV infected patients after DAA treatment [20, 22, 23]. While few have observed a partial or inexistent proliferative capacity recovery [24, 25]. Likewise, few and contradictory information have been reported about CD4+ T cell proliferative capacity after DAAs [24, 26].

Thus, there is still an open question on whether the elimination of HCV with DAAs results in a full, partial or inexistent restoration of the immune response.

In the present study, we investigate the phenotypical and functional changes of HCV-specific CD8+ and CD4+ T-cells responses in order to study the degree of immune restoration experienced by patients that achieved SVR after DAA treatment.

## Materials and methods

### Patients

Twenty-seven patients chronically infected with HCV were recruited from the Liver Disease Unit of Vall d'Hebron Hospital in Barcelona, Spain. The study was approved by the Institutional Review Board on Clinical Research of Vall d'Hebron Hospital (Code: HCV-SIR) and all subjects gave written informed consent in accordance with the 1975 Declaration of Helsinki.

Patient characteristics and clinical parameters are provided in Table 1 and the information of each patient is disclosed in S1 Table. All patients were HBV surface antigen and anti-HIV negative at the time of blood collection. The study included thirteen patients with HCV genotype 1a and fourteen patients with genotype 1b that received different combinations of IFN-

**Table 1. Patient's characteristics.**

| Patient characteristics (N = 27) | | |
|---|---|---|
| **Age, mean (±SD)** | 55 (±12.6) | |
| **Gender, N Male (%)** | 12 (44.4) | |
| **Liver stiffness, fibrosis score, N (%)** | | |
| < 9.5 KPa | 15 (55.5) | |
| ≥ 9.5 KPa | 12 (44.5) | |
| **HCV genotype 1 subtypes, N (%)** | | |
| 1a | 13 (48.1) | |
| 1b | 14 (51.9) | |
| **DAAs treatment, N (%)** | | |
| SOF/LDV [a] | 17 (63.0) | |
| SOF/LDV + RBV | 4 (14.8) | |
| SOF / SMV [b] | 1 (3.7) | |
| EBR / GZR + RBV [c] | 2 (7.4) | |
| OBV/PTV/R + DSV + RBV [d] | 3 (11.1) | |
| **Previous treatment, N (%)** | | |
| Naive | 16 (59.3) | |
| PEG-IFN / RBV | 9 (33.3) | |
| PEG-IFN / RBV / BOC [e] | 2 (7.4) | |
| **Transaminase enzymes, mean (±SD)** | **Baseline** | **FUW12** |
| ASG (UI/L) [f] | 60.5 (±38.7) | 23.9 (±8.4) |
| ALT (UI/L) [g] | 79.7 (±61.4) | 19.9 (±7.8) |
| Y-GT (UI/L) [h] | 68.6 (±62.8) | 21.6 (±7.9) |
| **Bilirubin, mean (± SD)** | **Baseline** | **FUW12** |
| Srm-bilirubin (ug/dL) [i] | 0.8 (±0.3) | 0.8 (±0.4) |
| Srm-esterified bilirubin (ug/dL) [j] | 0.3 (±0.1) | 0.3 (±0.1) |

[a] Sofosbuvir/Ledipasvir

[b] Sofosbuvir / Simeprevir

[c] Elbasvir / Grazoprevir

[d] Ombitasvir/Paritaprevir/Ritonavir+Dasabuvir + Ribavirin

[e] Pegylated Interferon-α / Boceprevir

[f] Aspartate aminotransferase. Normal values <35–50 UI/L

[g] Alanine aminotransferase. Normal values <50 UI/L

[h] Gamma glutamyl transferase. Normal values <55 UI/L.

[i] Total Bilirubin. Normal values <1.20 ug/dL

[j] Esterified bilirubin. Normal values <0.57 ug/dL

free DAA therapies. Eleven patients had received IFN-based regimens before DAA therapy. Fifteen patients were non-cirrhotic with mild fibrosis grade based on transient elastography (Liver stiffness < 9.5 KPa) and 12 patients had an advanced stage of liver fibrosis or cirrhosis (Liver stiffness > 9.5 KPa). Blood samples used for this study were collected at the following times: prior to DAA treatment (baseline), at 4 weeks of treatment (W4), at the end of the treatment (EOT), and 12 weeks after the end of treatment (FUW12). All treated individuals included in this study achieved a SVR (i.e., undetectable HCV RNA by 12 weeks after treatment cessation) and patients 7 and 9 developed HCC after 61 and 55 month of follow up respectively.

## Cell Isolation, cryopreservation and thawing

PBMCs were isolated from whole blood by density gradient centrifugation in BD Vacutainer-CPT Mononuclear Cell Preparation tubes (BD Biosciences, San Diego, CA). Samples were cryopreserved in medium containing 90% fetal calf serum (GIBCO BRL) and 10% DMSO (Sigma-Aldrich, St Louis, MO) and cryopreserved at -80˚C. All assays were performed with thawed PBMCs maintained in completed RPMI 1640 (GIBCO/Invitrogen) supplemented with 10% heat-inactivated human AB serum, 2 mmol/L L-glutamine and 100 μg/mL streptomycin.

## Phenotypic analysis of CD4+ T cell

After being thawed, $1x10^6$ PBMCs in completed medium were washed with PBS-0.5% fetal calf serum and stained with the fluorescently labeled anti-human antibodies CD4-FITC (BD Biosciences Cat#555346), PD1-PE (BD Biosciences Cat#557946), LAG3-BV421 (BD Biosciences Cat#565720), TIM3-AlexaFluor641 (BD Biosciences Cat#565558) and their corresponding isotype controls; FITC Mouse IgG1,k Isotype (BD Biosciences Cat#551954); PE Mouse IgG1,k Isotype (Beckman Coulter Cat#A07796), BV421 Mouse IgG1,k Isotype (BD Biosciences Cat#562438) and Alexa Fluor 647 Mouse IgG1 Isotype (BD Biosciences Cat#565571) at 4˚C, for 30 min in the dark. For the identification and quantitation of exhausted CD4+ T cell subsets, 100,000 events were acquired per sample and analyzed in a Fortessa equipped with the FACS DIVA software (BD Biosciences, San Jose, CA). BD FACS Express software (BD Biosciences, San Jose, CA) was used for data analysis. Results were expressed as a percentage of positive cells.

## HCV-specific CD4+ and CD8+ antigen-specific T cell responses

**HCV antigens for PBMC stimulation.** The 28 and 98 peptides (15–19-mers with 11–12 amino acid (aa) overlaps) spanning the core and nonstructural protein 3 (NS3), respectively, of HCV subtype 1a (H77) and 1b (J4) were obtained through the NIH Biodefense and Emerging Infectious Research Resources Repository, NIAID, NIH (peptide arrays, HCV Core and NS3 proteins, NR-3737, NR-3747, NR-3752, NR-37452). Purified recombinant HCV NS3-helicase (aa: 1207–1488) proteins derived from subtype 1a and 1b sequences and expressed in the yeast Pichia pastoris were purchased from Mikrogen (Neuried, Germany).

PBMCs were stimulated with 4 μg/mL of overlapping HCV Core or NS3 pool peptides or 2 μg/mL of NS3 Helicase protein subtype 1a or 1b. PBMCs stimulated with 1 μg/mL of Staphylococcus aureus enterotoxin B (SEB; Sigma-Aldrich, Deisenhofer, Germany) or incubated with completed medium alone were used as positive and negative controls, respectively.

**IFN-γ ELISpot assay.** Freshly thawed PBMCs were cultured at a density of $5x10^5$ cells per mL in completed medium and stimulated with HCV antigens as above. After overnight incubation at 37˚C and 5% $CO_2$, ELISPOT assays were performed according to previously published [27]. Results were expressed as the number of IFN-γ SFCs per $10^6$ PBMCs (IFN-γ

SFCs/$10^6$ PBMCs). Median number of IFN-γ SFCs/$10^6$ PBMCs stimulated with different HCV antigens was compared between the time points. Assays with high background or no SEB response were excluded.

**Intracellular cytokine staining.** Freshly thawed PBMCs at a density of 1x$10^6$ cells per ml in completed medium supplemented with 0.67 μL per ml of BD GolgiStop Protein Transport Inhibitor containing monensin (BD Biosciences, San Diego, CA) were stimulated with HCV antigens as described above and co-stimulated with 1 μg/ml of anti-human CD28 antibody (BD Biosciences, Heidelberg, Germany). After 12–16 h of incubation at 37°C and 5% $CO_2$, cells were stained with anti-human CD4-FITC (BD Biosciences Cat#555346), anti-human CD8-PE (BD Biosciences Cat#555635) or the respective isotype control antibodies for 30 min at 4°C in complete darkness, and then fixed and permeabilized with BD Cytofix/Cytoperm solution (BD Biosciences, San Diego, CA) [27]. After permeabilization, cells were stained with anti-human IFNγ-APC (BD Biosciences Cat#554702) and anti-human IL2-PerCP-Cy5.5 (BD Biosciences Cat#560708) or the respective isotype control antibodies: APC-Mouse IgG1,k Isotype (BD Biosciences Cat#550854) and PerCP-Cy5.5 Rat IgG2a Isotype (BD Biosciences Cat#550765) at 4°C for 30 min in the dark, and washed twice with 1xBD Perm/Wash Buffer. Flow cytometry data was acquired using a FACSCalibur (BD Biosciences, San Jose, CA) and analyzed with CellQuest software (BD Biosciences, San Jose, CA). The number of viable cells that had produced IFNγ and IL2 was determined by gating on the CD4+ and CD8+ positive for those two markers and substracting the background response.

**Proliferation assay of HCV-specific CD4+ and CD8+ antigen-specific T cell.** Antigen-specific T cell proliferation was determined in all patients by carboxy-fluorescein diacetate succinimidyl ester (CFSE) dilution assay as previously published [28]. After 5 days at 37°C, cells were washed and stained with anti-human CD4-PE (BD Biosciences Cat# 555347) and CD8-APC (BD Biosciences Cat# 555369) antibodies, and the viability dye 7-AAD. Flow cytometry analysis were performed on a FACSCalibur using CellQuest software (BD Biosciences, San Jose, CA). The number of viable cells that had proliferated was determined by gating on the CD4+CFSE$^{low}$ and CD8+CFSE$^{low}$. Proliferation index was calculated as the ratio between CD4+ or CD8+ proliferative frequency (%) in the presence of specific antigen and that in the absence of antigen, as previously reported [28].

## Statistical analysis

Results are presented as mean and SD. Comparisons between groups were performed with nonparametric Wilcoxon signed-rank test. P-values of less than 0.05 were considered significant. All Data was analyzed by GraphPad Prism 6.

## Results

### Partial reversion of the exhausted phenotype signatures in CD4+ T cells after DAA therapy depends on clinical parameters

Cell surface expression of Programmed cell death protein 1 (PD-1), T cell immunoglobulin and mucin domain-containing protein 3 (TIM-3), and Lymphocyte-activation gene 3 (LAG-3), which have been identified as markers of exhausted T cells, was tested by Flow Cytometry in baseline and FUW12 samples of the 27 patients. We observed a slight decrease of PD-1, TIM-3 and LAG-3 expression on CD4+ T cells in FUW12 samples compared to baseline samples, although the differences were not statistically different (Fig 1A). However, significant differences were observed when clinical parameters like fibrosis stage and previous IFN-based treatment were taken into account. By doing so, we observed that non-cirrhotic patients

showed a statistically significant decrease of PD-1 receptor expression at the surface of CD4
+ T cells from FUW12 samples compared to baseline (p = 0.033), while cirrhotic patients did
not show a clear tendency to reduce this T cell exhaustion marker (Fig 1B). Moreover, PD-1
surface expression in treatment-naive patients after HCV elimination at FUW12 was signifi-
cantly lower than that at the baseline (p = 0.010), while in those patients who received previous
IFN-α treatment, this decrease was not observed (Fig 1C).

In addition, only treatment-naive patients below 55 years of age showed a significant reduc-
tion of PD-1 expression (Fig 1D).

These data suggest that restoration of CD4+ T cells exhausted phenotype is achieved in
non-cirrhotic, treatment-naive and treatment-naïve below 55, while in cirrhotic, IFN-α previ-
ously treated and older age patients, HCV clearance does not fully revert the exhausted pheno-
type of CD4+ T cells.

### HCV specific CD4+ and CD8+ T cells do not reverse the impaired cytokine production after HCV elimination by DAA regimens

We assessed whether the impaired functionality of HCV-specific CD4+ and CD8+ T cells
manifested by reduced T helper 1 cytokine production (IL2 and IFN-γ) during chronic hepati-
tis C is restored upon HCV clearance with DAA treatments. To do so, PBMCs isolated from
blood samples of patients with chronic HCV infection at baseline, W4, EOT, and FUW12
were stimulated with HCV antigens as described in the methods section. Comparisons
between baseline and FUW12 samples of all patients included in the study did not reveal any
significant increase in IFN-γ production after HCV elimination (Fig 2A). However, the num-
ber of IFN-γ SFC per $10^6$ PBMCs significantly increased in FUW12 samples compared to base-
line in G1b patients after stimulation with NS3 Helicase (p = 0.042) or Core peptides
(p = 0.013). Moreover, a similar increase was observed when comparing FUW12 versus W4
sample-pairs after stimulation with Core peptides (p = 0.021), and FUW12 versus EOT sam-
ple-pairs after stimulation with NS3 31–44 peptides (p = 0.023) (Fig 2B).

To further study the immune response restoration status, intracellular cytokine levels pro-
duced by HCV-specific CD4+ and CD8+ T-cells after 16 hours of stimulation with HCV anti-
gens was determined in all patients at baseline, W4, EOT, and FUW12. The percentages of
CD4+ IFN-γ+ T cells and CD4+ IL2+ T cells at FUW12 were not significantly higher than that
observed in baseline or W4, independently of the HCV antigens used for stimulation (Fig 3A).
T cells stimulated with NS3 31–44 peptides showed a significant decrease in IL2 production in
CD4+ T cells between baseline or W4 and FUW12 samples (p = 0.014 and p = 0.043, respec-
tively) and in CD8+ T cells between W4 and EOT (p = 0.020), and between EOT and FUW12
(p = 0.040). Interestingly, a significant increase in IL2 production was observed in CD8+ T
cells after stimulation with Core peptides when comparing baseline with W4 (p = 0.028), EOT
(p = 0.031) and FUW12 (p = 0.029).

Our data indicate that cytokine production by HCV-specific CD4+ and CD8+ T cells
remains impaired after HCV elimination by DAA treatments. However IL2 production
increases in CD8+ T cells stimulated with core antigen.

### Proliferative capacity on HCV-specific CD4+ and CD8+ T cells is restored in the majority of chronic hepatitis C patients after HCV elimination with DAA treatments

To determine whether the limited proliferation capacity of HCV-specific CD4+ and CD8+ T
cells observed during chronic HCV infection was reversed following the elimination of the
virus with DAA therapy, the proliferative capacity of these cells before and after DAA therapy

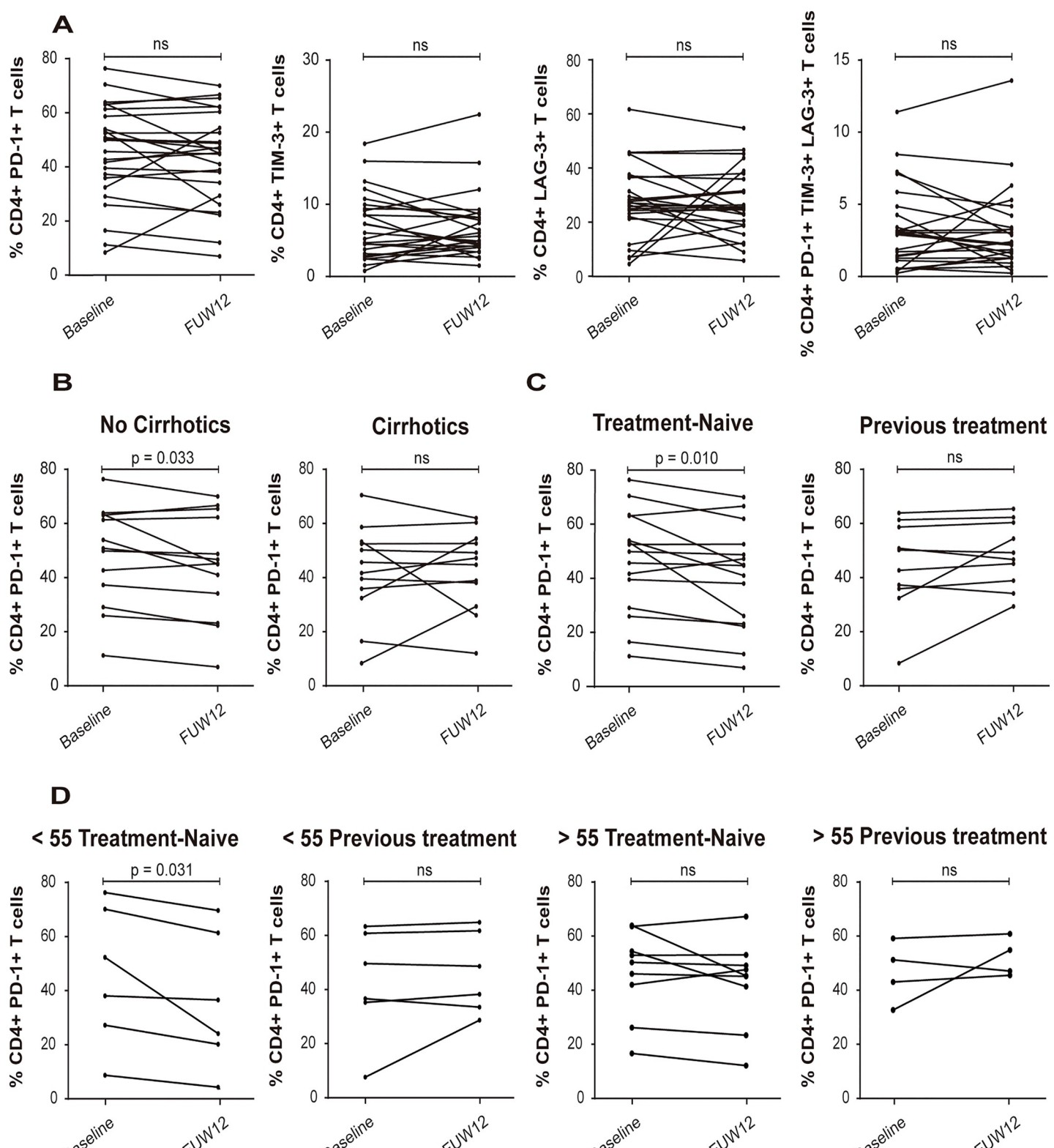

**Fig 1. CD4+ T cell exhaustion markers expression.** (A) Analysis of the frequencies of PD-1, TIM-3, LAG-3 markers among CD4+ T cells at baseline and FUW12, (N = 25). (B) Percent of PD-1 among CD4+ T cells according to score fibrosis. Non cirrhotic patients (N = 13), liver stiffness < 9.5 KPa; Cirrhotic patients (N = 12), liver stiffness ≥ 9.5 KPa. (C) Percent of CD4+ PD-1+ T cells in treatment-naive patients (N = 15), and in patients with previous IFN-based therapy (PEG-IFN-α + RBV) (N = 10). (D) Percent of CD4+ PD-1+ T cells in treatment-naive patients below 55 (N = 6), in patients below 55 with previous PEG-IFN-α + RBV treatment (N = 6), in

treatment-naive patients over 55 (N = 9) and in patients below 55 previously treated with PEG-IFN-α + RBV (N = 4). Statistical significance was determined by nonparametric Wilcoxon signed-rank test and represented by p-value. P < 0.05 was considered statistically significant.

was compared. A significant increase in the proliferative capacity of both CD4+ and CD8+ T cells between baseline and FUW12, after stimulation with NS3 Helicase (p = 0.049 and p = 0.012, CD4+ and CD8+ T cells, respectively), NS3 31–44 pooled peptides (p = 0.031 and p = 0.028, CD4+ and CD8+ T cells, respectively) and NS3 45–61 pooled peptides (p = 0.004 and p = 0.012, CD4+ and CD8+ T cells, respectively) was observed (Fig 4A). On the other hand, when considering fibrosis score, proliferation capacity of CD4+ and CD8+ T cells significantly increased in non-cirrhotic patients after NS3 31–44 peptides stimulation compared with cirrhotic ones (p = 0.034 and p = 0.041, respectively) (Fig 4B).

In summary, our data suggest that DAAs-mediated HCV clearance partially restores the proliferative capacity of virus-specific CD4+ and CD8+ T cells.

## Liver inflammation and fibrosis markers

To determine if liver inflammation and fibrosis stage regression occurred after therapy-mediated HCV elimination, liver inflammation and fibrosis markers were measured at baseline and FUW12 for each patient. All markers including transaminases, Fibrosis-4 (FIB-4) index, α-fetoprotein (AFP), and AST-to-platelet ratio index (APRI) showed a significant reduction (p = < 0.0001) 12 weeks after DAA-treatment (Fig 5). These data indicate that a reduction in liver inflammation and a fibrosis stage regression was achieved in all patients after DAA regimens, suggesting that a liver regeneration was still ongoing.

## Discussion

Persistent antigen stimulation, as in chronic infections, leads to T cell exhaustion and dysfunction, which results in an impaired immune response against the virus [29]. The exhaustion of HCV-specific T cells is characterized by up-regulation of PD-1 and other inhibitory receptors, low proliferative capacity, dysfunctional CD8 cytotoxicity, and impaired production of immunomodulatory cytokines [30–32].

In the present study we evaluated the degree of immune restoration by performing the analysis of phenotypic and functional changes in T cells before, during and after HCV elimination by DAA therapies.

Because host factors, including age and gender, can largely influence the immune response, the study cohort included both males and females in their middle age and elderly patients. Most patients with advanced liver fibrosis had abnormal baseline levels of liver inflammation and fibrosis markers, including transaminases, FIB-4, AFP, and APRI [33, 34]. All patients achieved SVR irrespective of the specific DAA treatment. HCV clearance was associated with decreased levels of liver inflammation and fibrosis markers, which correlates with the reduction of liver fibrosis related cytokines after HCV eradication by DAA treatment observed by Sasaki et al. [35].

Recent studies have demonstrated that DAA treatments reduce but do not eliminate the risk of developing HCC [36]. Indeed, 2 out of 27 patients who received DAA regimens in this study developed HCC at months 54 and 60 of follow-up, corresponding to a 5-years cumulative HCC rate of 7.4%. Despite the lack of specific predictors of HCC, several host factors, including liver fibrosis stage, older age, elevated AFP levels, and comorbidities such as diabetes and steatosis, have been associated with HCC occurrence [33, 37, 38]. Remarkably, patients who developed HCC were the oldest in our patients' cohort, suggesting that older age along

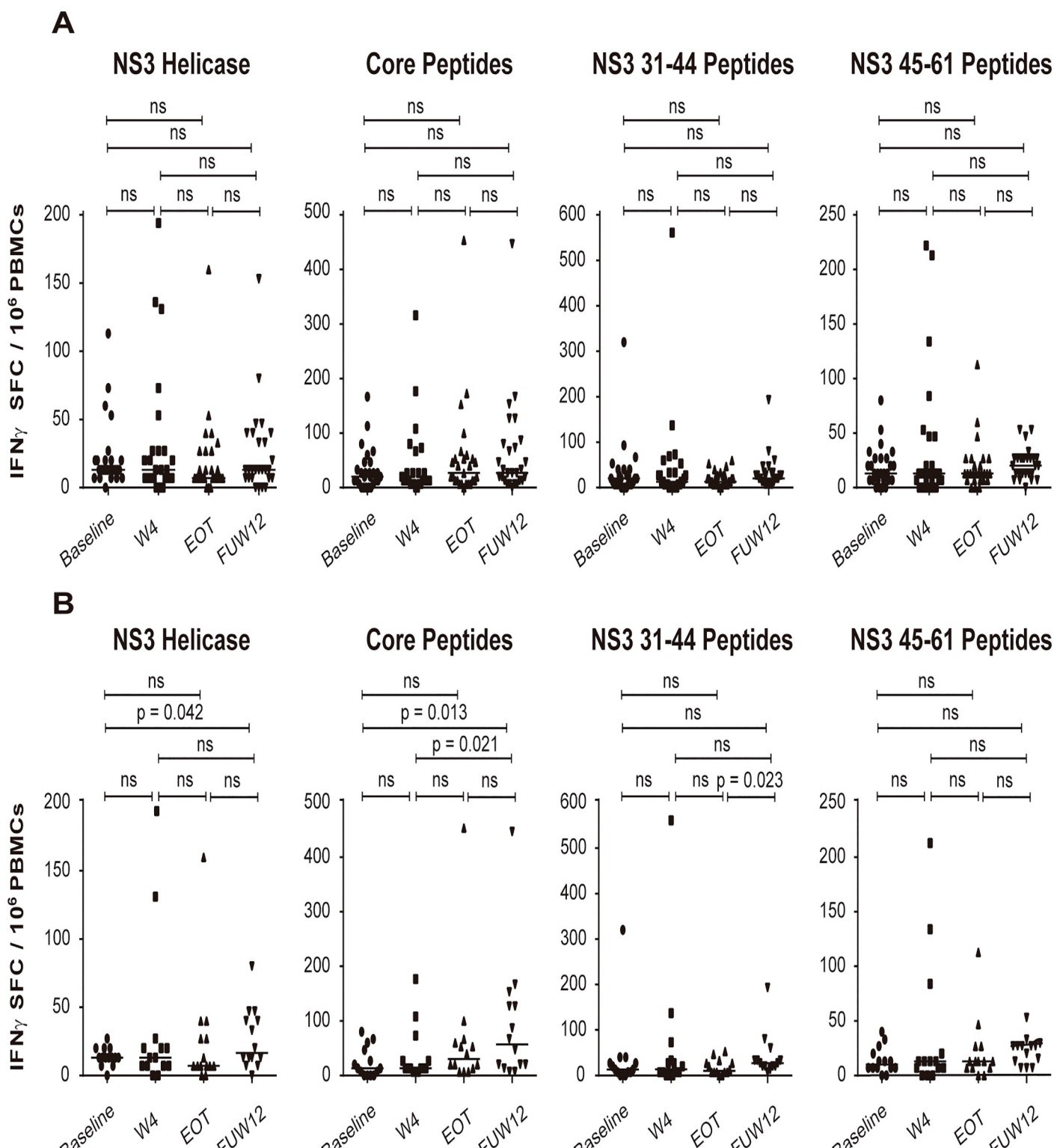

**Fig 2. HCV specific T cells immune response analyzed by IFN-γ ELISPOT.** Results were expressed as number of SFC producing IFN-γ. (A) IFN-γ production of $10^6$ PBMCs tested at baseline, W4, EOT, and FUW12. Total PBMCs where stimulated with HCV Peptides (NS3 Helicase, Core Peptides, NS3 31–44 Peptides and NS3 45–61 Peptides), (N = 27). (B) IFN-γ production of $10^6$ PBMCs tested in subtype 1b patients (N = 14). Horizontal bars represent median of IFN-γ SFC and each dot represents IFN-γ response from one patient. Median of IFN-γ SFC between different times were compared with Wilcoxon signed-rank test. P < 0.05 was considered statistically significant.

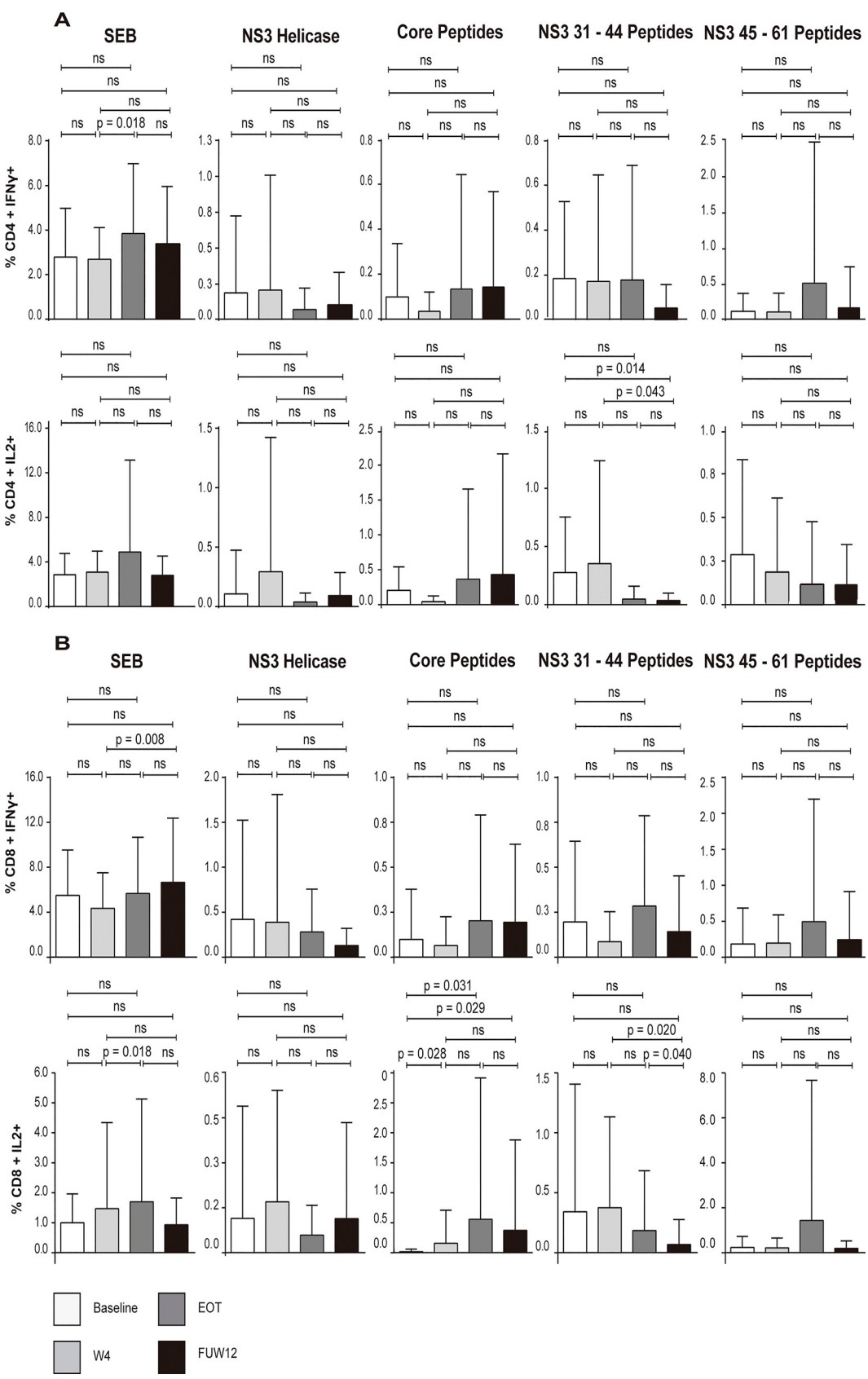

**Fig 3. Intracellular cytokines staining (IFN-γ and IL-2) secreted by T cells of HCV-infected individuals.** Frequencies of IFN-γ and IL-2 producing CD4+ and CD8+ specific T-cells were tested at 4 time points; baseline, W4, EOT and FUW12. T cells were stimulated with anti-CD28, monesin-containing transport inhibitor, HCV Peptides (NS3 Helicase, Core Peptides, NS3 31–44 Peptides, NS3 45–61 Peptides) and SEB as a positive control. (A) Percent CD4+ T cells producing IFN-γ and IL-2, (N = 22). (B) Percent CD8+ T cells producing IFN-γ and IL-2, (N = 22). Statistical significance was determined by Wilcoxon signed-rank test. P < 0.05 was considered statistically significant.

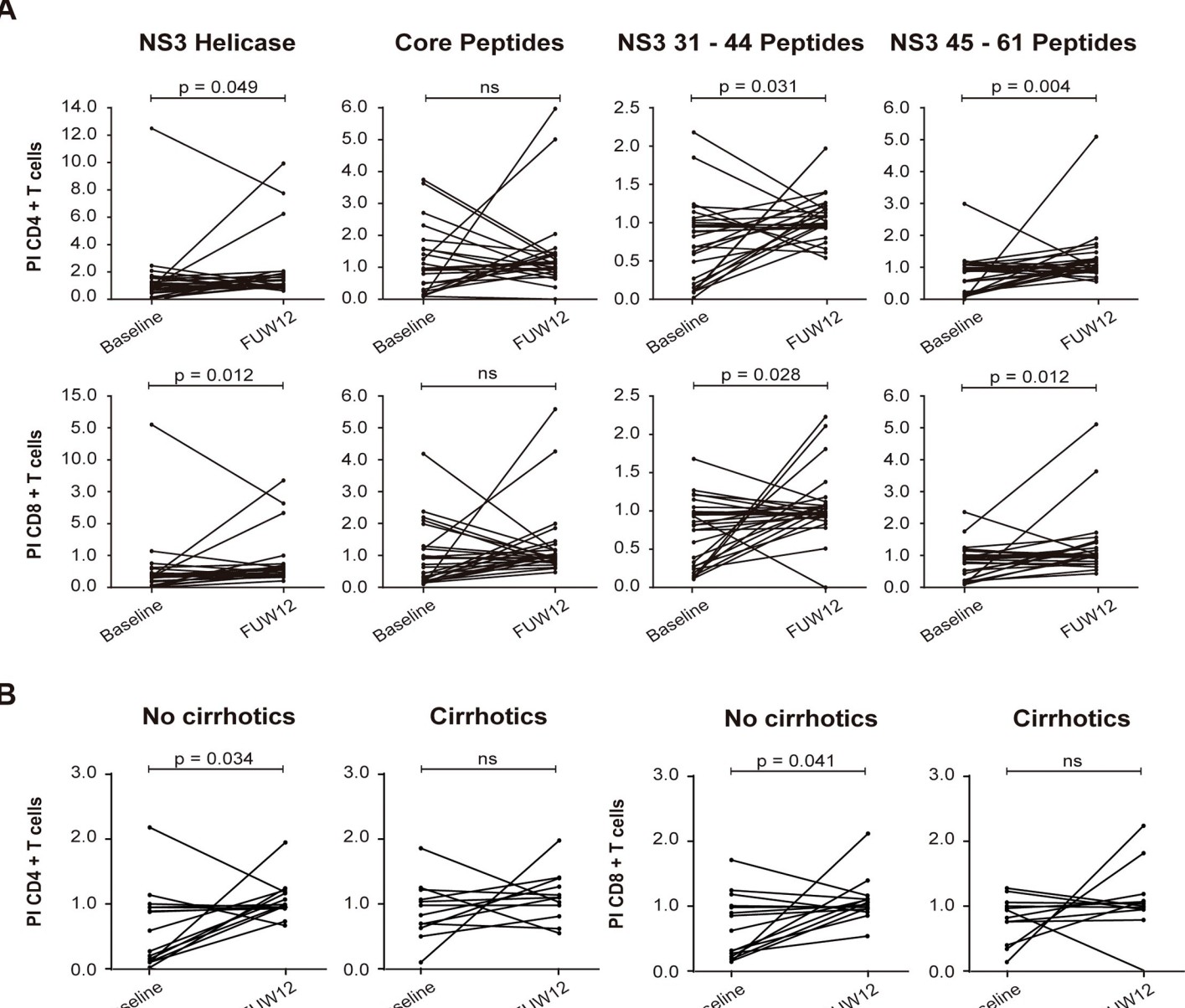

**Fig 4. Proliferation responses of CD4+ and CD8+ HCV specific T-cells.** In vitro expansion of PBMCs after 5 days of stimulation with HCV peptides (NS3 Helicase, Core Peptides, NS3 31–44 Peptides, NS3 45–61 Peptides). Proliferation capacity was calculated as the proliferation index, at baseline and FUW12, (A) Proliferation index of CD4+ and CD8+ T cells (N = 27). (B) T cells proliferation index according to score fibrosis when stimulated with NS3 31–44 Peptides. CD4+ and CD8+ T cells of non-cirrhotic patients (N = 15) and (N = 12) respectively, liver stiffness < 9.5 KPa; CD4+ and CD8+ T cells of cirrhotic patients (N = 12) and (N = 12) respectively, liver stiffness ≥ 9.5 KPa. Statistical significance was determined by nonparametric Wilcoxon signed-rank test and represented by p-value. P < 0.05 was considered statistically significant.

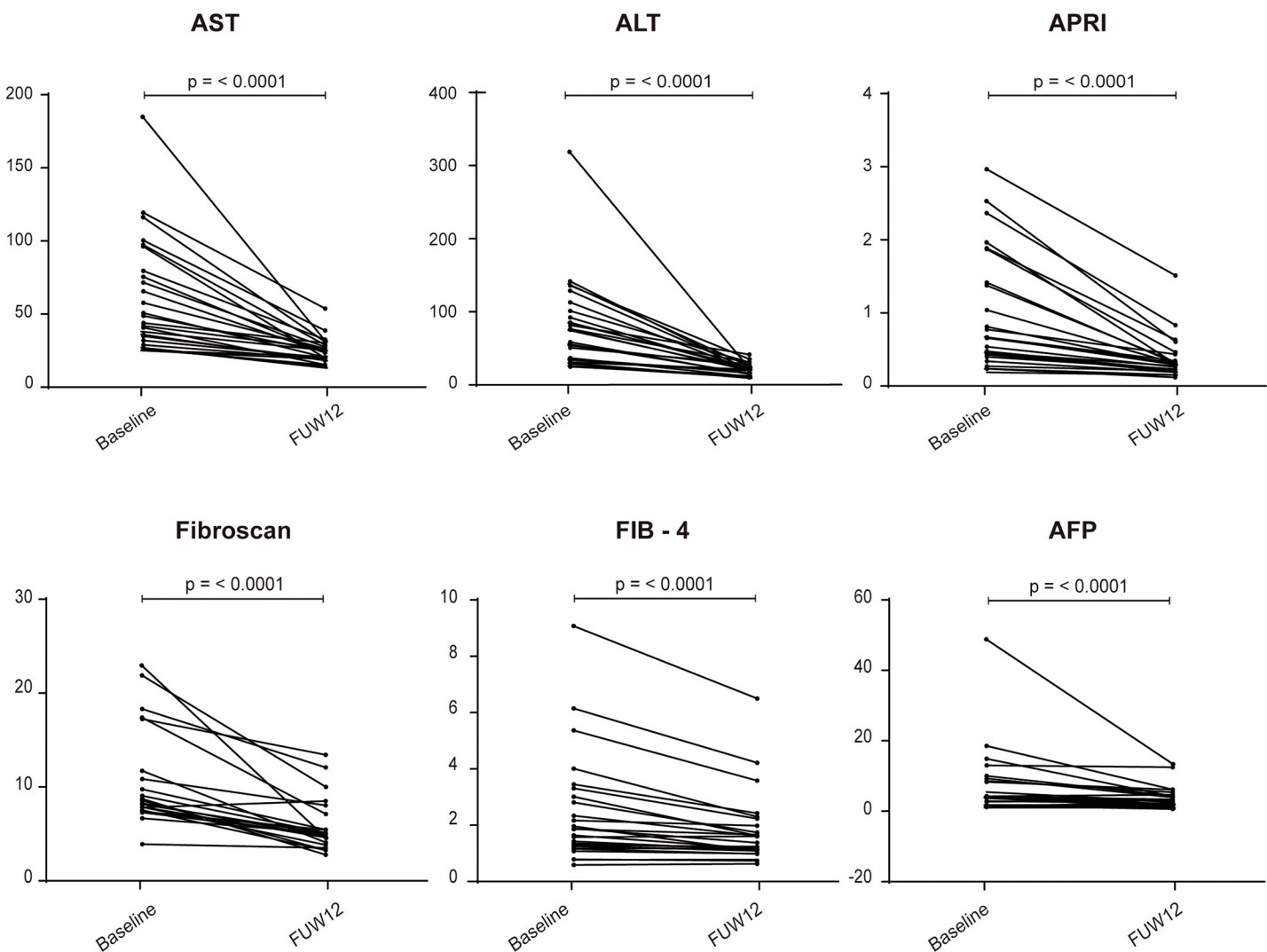

**Fig 5. Liver inflammation and fibrosis markers.** Levels of transaminases (AST, ALT), APRI, fibroscan, FIB-4 and AFP were measured at baseline and FUW12. Statistical significance was determined by Wilcoxon signed-rank test. P < 0.05 was considered statistically significant.

with advanced fibrosis stage at baseline was a significant driving factor in the HCC development in these patients. Other host factors that may have contributed to the occurrence of HCC include the elevated AFP after DAA treatment in patient 7 and diabetes in patient 9.

To study the restoration of immune response after DAA treatment, we first characterized the exhausted phenotype of CD4+ T cells before and FUW12 by analyzing the expression of immune response inhibitory markers PD-1, TIM-3, and LAG-3 at the cell surface.

Interestingly, a significant decrease in cell surface expression of PD-1 was observed in FUW12 samples compared with its baseline pair in non-cirrhotic and treatment-naive patients, suggesting that the immune response in these patients might have partly recovered. However, no significant differences were found when considering cirrhotic and IFN-α treated patients, which is in concordance with previous studies [25, 39], indicating that CD4 exhausted phenotype is not fully restored in all DAA-treated patients resolving infection.

The fact that after HCV eradication, PD1 expression levels were significantly reduced in treatment-naive patients compared with IFN-α treated ones and in non-cirrhotic patients

compared with cirrhotic ones, suggests that cirrhotic's derived T cells show a deeper functional exhaustion than the non-cirrhotic ones, which may be associated with the difference in liver damage and more impaired HCV-specific immune response. Moreover, it is possible that treatment-naive patients might be more receptive to respond to new treatment than IFN-α treated ones, as IFN-based therapies could have induced phenotypical changes towards T cells, making them to be less prone to switch off the exhausted phenotype. In addition, no significant differences in PD-1 levels at baseline and FUW12 were observed among treatment-naive patients over 55 years of age, indicating that older patients may be less likely to reverse T cell exhaustion. Thus, failure to reverse the exhausted phenotype may have also contribute to HCC development in patients 7 and 9.

In summary, a partial restoration of the CD4+ T cell exhausted phenotype in patients with low grade of fibrosis, in treatment-naïve and in treatment-naïve patients below 55 years old, was observed.

Regarding the pro-inflammatory cytokines, it has been reported that NK and T cells produce high quantities of IFN-γ and other pro-inflammatory cytokines during acute infections, while production is decreased during chronic infections [24, 40, 41].

Interestingly, Sasaki et al. reported higher levels of pro-inflammatory cytokines, including IFN- γ and IL-2, in serum of rapid virological responders as compared with end of treatment responders, suggesting that enhanced host immune status may contribute in HCV clearance [35]. Here, we have measured IFN-γ and IL-2 production of lymphocytes stimulated in vitro, comparing the baseline and FUW12 samples of each patient and found that HCV clearance per se is not sufficient to reverse the decreased production of IFN-γ and IL-2 by HCV-specific T cells, suggesting that prolonged exposure to HCV antigens leads to long-lasting HCV-specific T cells that lack effector functions. Thus, we could show that impaired HCV-specific CD4 + and CD8+ T cells responses during chronic HCV infection are not restored following successful HCV clearance, at least in the short-time after HCV elimination, which is in line with recent published studies [25, 39, 41–43].

During chronic infections, virus-induced transcriptional reprogramming contributes to the maintenance of exhausted T cells in hyporesponsive states [8, 29, 44, 45]. As we have shown, T cell functionality remain impaired despite SVR, suggesting persistent transcription factor or epigenetic changes in HCV-specific T-cell after infection resolution. Moreover, those alterations have been reported to be directly associated with advanced liver diseases as HCC [46, 47].

The analysis of IFN-γ ELISPOT results by HCV subtypes revealed that patients with HCV subtype 1b showed higher IFN-γ restoration than those with HCV subtype 1a, which could be attributed to a different immunogenicity between subtypes [48], or due to T-cell-mediated protective immunity against other viral strains to which the patient was previously exposed [49].

Various studies point to the fact that a restoration of HCV-specific CD8+ T cells proliferation after DAAs treatment occurs [20, 22, 23]. But others reported that CD8 proliferation capacity was not restored after HCV elimination by DAAs in the majority of patients [24, 25]. However, little data was reported about the proliferative capacity of HCV-specific CD4+ T cells until now. A recently published study showed limited CD4+ T cell proliferative capacity of HCV specific CD4+ T-cells following DAA therapy [26]. In our study, we have observed a partial restoration of the proliferative capacity of CD4+ and CD8+ T cells after stimulation with NS3 Helicase protein and NS3 pooled peptides but not with structural HCV antigens (core), suggesting that restoration of T cell proliferation is HCV-epitope dependent, as core antigen has been related with the induction of T regs expansion and T cell exhaustion [50, 51] and non-structural antigens have been described to present a higher immunogenicity [28, 52,

53]. In accordance to our results, several studies confirmed a reinvigorated CD8+ T cell proliferative capacity after DAAs when stimulated with NS3 and NS4 peptides [20, 22, 23]. In addition, Burchill et al. also found a temporal increase in the proliferative response of CD4+ T cells when stimulating with NS3 and NS5 [24].

The discrepancies between our results and other studies pointing to a non-restoration of the proliferative capacity could be explained by different clinic-pathological conditions, by different DAA regimen used or due to the differences on immunogenicity between peptides.

Overall, our data suggest that clearance of persistent HCV antigens helps to partially increase the proliferative capacity of CD4+ and CD8+ T cells but is not sufficient to reverse T cell dysfunctionality.

The fact that those T cells remain dysfunctional after HCV clearance might have serious clinical implications. Additional immunological therapy may reduce the risk of developing HCC and other extrahepatic manifestations after DAA regimen [54]. Unfortunately, patients who have overcome the infection and cleared the virus are not protected against reinfection [55]. Moreover, no vaccine to prevent HCV infection is yet available, despite continuing efforts for its development [56]. Our study will provide information on the dynamics of HCV infection and immune response, which may be applicable to vaccine development and immunotherapy strategies, as well as on immune related liver remodeling that together with a correct diagnose and an early treatment will help to improve clinical outcomes.

Since this study has been limited to a low number of patients, similar studies with a higher number of patients need to be performed in order to avoid missing any biological significant trend. In addition, another limiting factor consists in the high intra-individual variability. Hence, we have designed the study so that each patient was their own control and comparisons were done intra-patient, avoiding additional genetic variations.

In conclusion, our study demonstrates a partial restoration of HCV-specific immune response in chronic HCV infected patients after the viral clearance induced by DAAs. Further investigations of HCV-specific T cell responses beyond 1 year of follow-up are needed to understand the long-term impact of HCV cure and their implications on HCV vaccine-design and HCV re-infections.

## Supporting information

**S1 Table. Clinical parameters of each patient.** Age, gender, HCV genotype, previous Treatment, DAA regimens, fibroscan, transaminase enzymes (AST, ALT), APRI, FIB-4 and AFP levels were taken at baseline and FUW12.
(PDF)

## Acknowledgments

The authors thank the Department of External extractions of Vall d'Hebron University Hospital (VHUH), especially to Francisco Tena who performed sample extractions and Judit Carbonell, who made the liver stiffness. Also thank Banc de Sang i Teixits (BST), Vall d'Hebron Institut de Recerca (VHIR), VHUH and Unitat d'Alta Tecnologia (UAT).

## Author Contributions

**Conceptualization:** Maria Isabel Costafreda, Celia Perales, Silvia Sauleda, Juan Ignacio Esteban, Marta Bes.

**Data curation:** Meritxell Llorens-Revull, Mercedes Guerrero-Murillo, Juan Ignacio Esteban.

**Formal analysis:** Meritxell Llorens-Revull, Angie Rico, Maria Eugenia Soria, Pablo Gabriel-Medina, Francisco Rodríguez-Frías, Marta Bes.

**Funding acquisition:** Celia Perales, Josep Quer, Silvia Sauleda, Juan Ignacio Esteban, Marta Bes.

**Investigation:** Meritxell Llorens-Revull, Maria Isabel Costafreda, Angie Rico, Sofía Píriz-Ruzo, Elena Vargas-Accarino, Mar Riveiro-Barciela, Marta Bes.

**Methodology:** Meritxell Llorens-Revull, Maria Isabel Costafreda, Maria Eugenia Soria, Elena Vargas-Accarino, Pablo Gabriel-Medina, Francisco Rodríguez-Frías, Celia Perales, Marta Bes.

**Project administration:** Marta Bes.

**Resources:** Celia Perales, Juan Ignacio Esteban, Marta Bes.

**Software:** Mercedes Guerrero-Murillo.

**Supervision:** Maria Isabel Costafreda, Marta Bes.

**Validation:** Mar Riveiro-Barciela, Celia Perales, Marta Bes.

**Writing – original draft:** Meritxell Llorens-Revull.

**Writing – review & editing:** Maria Isabel Costafreda, Maria Eugenia Soria, Mar Riveiro-Barciela, Celia Perales, Josep Quer, Silvia Sauleda, Juan Ignacio Esteban, Marta Bes.

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
