## [Decision Letter · Decision Letter 0]

26 Apr 2021

PONE-D-21-11447

Partial restoration of immune response in Hepatitis C patients after viral clearance by direct-acting antiviral therapy

PLOS ONE

Dear Dr. Marta Bes,

Thank you for submitting your manuscript to PLOS ONE. After careful consideration, we feel that it has merit but does not fully meet PLOS ONE’s publication criteria as it currently stands. Therefore, we invite you to submit a revised version of the manuscript that addresses the points raised during the review process.

We look forward to receiving your revised manuscript.

Kind regards,

Tatsuo Kanda, M.D., Ph.D.

Academic Editor

PLOS ONE

Journal Requirements:

We note that you have included the phrase “data not shown” in your manuscript. Unfortunately, this does not meet our data sharing requirements. PLOS does not permit references to inaccessible data. We require that authors provide all relevant data within the paper, Supporting Information files, or in an acceptable, public repository. Please add a citation to support this phrase or upload the data that corresponds with these findings to a stable repository (such as Figshare or Dryad) and provide and URLs, DOIs, or accession numbers that may be used to access these data. Or, if the data are not a core part of the research being presented in your study, we ask that you remove the phrase that refers to these data.

Reviewers' comments:

Reviewer's Responses to Questions

**Comments to the Author**

1. Is the manuscript technically sound, and do the data support the conclusions?

Reviewer #1: Partly

Reviewer #2: Yes

2. Has the statistical analysis been performed appropriately and rigorously? 

Reviewer #1: I Don't Know

Reviewer #2: Yes

3. Have the authors made all data underlying the findings in their manuscript fully available?

Reviewer #1: No

Reviewer #2: Yes

4. Is the manuscript presented in an intelligible fashion and written in standard English?

Reviewer #1: Yes

Reviewer #2: Yes

5. Review Comments to the Author

Reviewer #1: Meritxell Llorens-Revull et al. reported that, in chronically infected patients, HCV elimination by DAA treatment leads to partial restoration of HCV-specific immune responses. They also mentioned that proliferative capacity of HCV-specific CD4+ and CD8+ T cells is recovered after DAA’s therapies, although they did not examine those from the controls.

1. In abstract section, authors used “HCV CD8+ and CD4+ specific T-cells”. What does it mean “HCV specific-immune response”??

2. Authors include HCV GT1a and GT1b patients. Authors used several different regimens of DAA treatments. Is there any difference?

3. “HCV-specific 154 CD4+ and CD8+ antigen-specific T cell responses” The 28 and 98 peptides (15–19-mers with 11–12 amino acid (aa) overlaps) spanning the core and nonstructural protein 3 (NS3), respectively, of HCV subtype 1a and 1b were obtained through the NIH Biodefense and Emerging Infectious Research Resources Repository, NIAID, NIH (peptide arrays, HCV, H77, core and NS3 proteins, NR-3737, NR-3747, NR-3752, NR-37452). H77 is a HCV GT1a strain. What strain did you use for HCVGT1b?

4. In Discussion section, authors should refer the following reference: Sasaki R, et al. Rapid hepatitis C virus clearance by antivirals correlates with immune status of infected patients. J Med Virol. 2019 Mar;91(3):411-418. doi: 10.1002/jmv.25310. PMID: 30192392

Please discuss more and mention the different findings from them.

5. Authors should describe clinical courses of all patients in detail.

Reviewer #2: The authors, Meritxell Llorens-Revull. et al., report partial restoration of immune response in Hepatitis C patients after viral clearance by direct-acting antiviral therapy.

They analyzed proliferative capacity of HCV-specific CD4+ and CD8+ using cell surface expression of Programmed cell death protein 1 (PD-1), T cell immunoglobulin and mucin domain-containing protein 3 (TIM-3), and Lymphocyte-activation gene 3 (LAG-3), which have been identified as markers of exhausted T cells.

It's a good point of view to focus on the recovery of the immune response, not just the elimination of the virus.

DAA eliminate the virus at a high rate, so, this study needs to focus on how it relates to carcinogenesis.

Major comments:

1. The authors should add, AFP,(L1, L3), DCP, ATX, M2BPGi as a carcinogenesis marker and fibrosis marker the point of baseline, W4, EOT and FUW12.

2. If possible, indicate the status of subsequent cancer rates Due to the difference in partial restoration of immune response.

3. The immune system deteriorates with age, but how does this compare with age?

Minor comments:

1.The author should confirm Table1 for confusion between product names and generic names.

6. PLOS authors have the option to publish the peer review history of their article (what does this mean?). If published, this will include your full peer review and any attached files.

Reviewer #1: No

Reviewer #2: **Yes: **Hiroteru Kamimura

---

## [Author Response · Author response to Decision Letter 0]

9 Jun 2021

Journal Requirements:

Answer: We have adjusted the manuscript to meet PLOS ONE style requirements.

Answer: We have eliminated the phases with “data not shown” or included the data in the text.

3. Have the authors made all data underlying the findings in their manuscript fully available?

The PLOS Data policy requires authors to make all data underlying the findings described in their manuscript fully available without restriction, with rare exception (please refer to the Data Availability Statement in the manuscript PDF file). The data should be provided as part of the manuscript or its supporting information, or deposited to a public repository. For example, in addition to summary statistics, the data points behind means, medians and variance measures should be available. If there are restrictions on publicly sharing data e.g. participant privacy or use of data from a third party those must be specified.

Answer to points 2 and 3: All relevant data underlying the findings have been included as Figure 4B, lines 340-344.

4. Review Comments to the Author

Reviewer #1: Meritxell Llorens-Revull et al. reported that, in chronically infected patients, HCV elimination by DAA treatment leads to partial restoration of HCV-specific immune responses. They also mentioned that proliferative capacity of HCV-specific CD4+ and CD8+ T cells is recovered after DAA’s therapies, although they did not examine those from the controls.

1. In abstract section, authors used “HCV CD8+ and CD4+ specific T-cells”. What does it mean “HCV specific-immune response”?

Answer: We apologize for the lack of clarity. Our results demonstrated that DAA treatment lead to the partial reversion of CD4+ T cell exhaustion and the restoration of the proliferative capacity of HCV-specific CD4+ and CD8+ T cells. Then, we have edited the abstract section to explain these results more clearly as “… to partial reversion of CD4+ T cell exhaustion.” See line 57 of the clean version.

2. Authors include HCV GT1a and GT1b patients. Authors used several different regimens of DAA treatments. Is there any difference?

Answer: We thank reviewer#1 for raising this question. We have included the information in lines 390-391 in the clean version: “All patients achieved SVR irrespective of the specific DAA treatment.” Unfortunately, statistical analysis between different DAA treatments cannot be performed due to the small number of patients treated with DAAs other than sofosbuvir+ledipasvir.

3. “HCV-specific 154 CD4+ and CD8+ antigen-specific T cell responses” The 28 and 98 peptides (15–19-mers with 11–12 amino acid (aa) overlaps) spanning the core and nonstructural protein 3 (NS3), respectively, of HCV subtype 1a and 1b were obtained through the NIH Biodefense and Emerging Infectious Research Resources Repository, NIAID, NIH (peptide arrays, HCV, H77, core and NS3 proteins, NR-3737, NR-3747, NR-3752, NR-37452). H77 is a HCV GT1a strain. What strain did you use for HCVGT1b?

Answer: We have added the required information in line 170 of the clean version. The HCVGT1b strain used was J4.

4. In Discussion section, authors should refer the following reference: Sasaki R, et al. Rapid hepatitis C virus clearance by antivirals correlates with immune status of infected patients. J Med Virol. 2019 Mar; 91(3):411-418. doi: 10.1002/jmv.25310. PMID: 30192392. Please discuss more and mention the different findings from them. 

Answer: We thank the reviewer#1 for this comment. We have found this reference very interesting and commented the results in the discussion. See lines 391-394 “HCV clearance was associated with decreased levels of liver inflammation and fibrosis markers, which correlates with the reduction of liver fibrosis related cytokines after HCV eradication by DAA treatment observed by Sasaki et al(35)” and also lines 441-444 “Interestingly, Sasaki et al. reported higher levels of pro-inflammatory cytokines, including IFN- Ɣ and IL-2, in serum of rapid virological responders as compared with end of treatment responders, suggesting that enhanced host immune status may contribute in HCV clearance(35)” 

5. Authors should describe clinical courses of all patients in detail.

Answer: We have added new data related to liver inflammation and fibrosis. In addition, we have added a new Table as supplementary information (S1 Table), which includes a comprehensive overview of the clinical parameters at baseline and FUW12 of all patients. Quoted in line 115.

Reviewer #2: The authors, Meritxell Llorens-Revull. et al., report partial restoration of immune response in Hepatitis C patients after viral clearance by direct-acting antiviral therapy.

They analyzed proliferative capacity of HCV-specific CD4+ and CD8+ using cell surface expression of Programmed cell death protein 1 (PD-1), T cell immunoglobulin and mucin domain-containing protein 3 (TIM-3), and Lymphocyte-activation gene 3 (LAG-3), which have been identified as markers of exhausted T cells.

It's a good point of view to focus on the recovery of the immune response, not just the elimination of the virus.

DAA eliminate the virus at a high rate, so, this study needs to focus on how it relates to carcinogenesis.

Major comments:

1. The authors should add, AFP, (L1, L3), DCP, ATX, M2BPGi as a carcinogenesis marker and fibrosis marker the point of baseline, W4, EOT and FUW12. 

Answer: We thank the reviewer#2 for this comment that significantly improved results and enriched our discussion. We have made an effort to report the clinical and biochemical analytical parameters suggested by reviewer #2. We have measured the AFP levels at baseline and FUW12 and include results of other markers of liver inflammation and fibrosis, such as Fibrosis-4 (FIB-4) index, transaminases, and AST-to-platelet ratio index (APRI). Results are shown in figure 5, described in lines 361-369, and values are shown as S1 Table (supplementary material). The added results have enriched discussion focussing in carcinogenesis. See discussion lines 388-394, 398-401 and 404-406.

2. If possible, indicate the status of subsequent cancer rates Due to the difference in partial restoration of immune response. 

Answer: We thank reviewer#2 for raising up this interesting question. Of note, we have found 2 patients that recently developed hepatocellular carcinoma (HCC). This is a very important finding because none of our patients developed HCC after 50 months of follow-up, however, when re-assessing the clinical outcomes of each patient to answer reviewer’s questions, the cumulative HCC rate after 60 months of follow-up had increased up to 7.4%. See material and methods section; lines 127-128 and discussion section; lines 396-398.

3. The immune system deteriorates with age, but how does this compare with age? 

Answer: This is a very interesting question that is now considered in the results and discussion. We have found that age has a significant impact on the capacity to reverse CD4 T cell exhausted phenotype. See Figure 1D and lines 252-253, 268-271, 398-404, lines 426-431 and lines 434-435 in discussion section. 

Minor comments:

1. The author should confirm Table1 for confusion between product names and generic names. 

Answer: Table 1 has been corrected, removing all product names and referring generic names.

---

## [Decision Letter · Decision Letter 1]

23 Jun 2021

Partial restoration of immune response in Hepatitis C patients after viral clearance by direct-acting antiviral therapy

PONE-D-21-11447R1

Dear Dr. Marta Bes,

We’re pleased to inform you that your manuscript has been judged scientifically suitable for publication and will be formally accepted for publication once it meets all outstanding technical requirements.

Kind regards,

Tatsuo Kanda, M.D., Ph.D.

Academic Editor

PLOS ONE

Additional Editor Comments (optional):

Reviewers' comments:

Reviewer's Responses to Questions

**Comments to the Author**

1. If the authors have adequately addressed your comments raised in a previous round of review and you feel that this manuscript is now acceptable for publication, you may indicate that here to bypass the “Comments to the Author” section, enter your conflict of interest statement in the “Confidential to Editor” section, and submit your "Accept" recommendation.

Reviewer #1: All comments have been addressed

2. Is the manuscript technically sound, and do the data support the conclusions?

Reviewer #1: Yes

3. Has the statistical analysis been performed appropriately and rigorously? 

Reviewer #1: I Don't Know

4. Have the authors made all data underlying the findings in their manuscript fully available?

Reviewer #1: Yes

5. Is the manuscript presented in an intelligible fashion and written in standard English?

Reviewer #1: Yes

6. Review Comments to the Author

Reviewer #1: (No Response)

7. PLOS authors have the option to publish the peer review history of their article (what does this mean?). If published, this will include your full peer review and any attached files.

Reviewer #1: No

---

## [Editor Report · Acceptance letter]

1 Jul 2021

PONE-D-21-11447R1 

Partial restoration of immune response in Hepatitis C patients after viral clearance by direct-acting antiviral therapy 

Dear Dr. Bes:

I'm pleased to inform you that your manuscript has been deemed suitable for publication in PLOS ONE. Congratulations! Your manuscript is now with our production department. 

Kind regards, 

on behalf of

Dr. Tatsuo Kanda 

Academic Editor

PLOS ONE